# Estimating Low- and High-Cyclic Fatigue of Polyimide-CF-PTFE Composite through Variation of Mechanical Hysteresis Loops

**DOI:** 10.3390/ma15134656

**Published:** 2022-07-02

**Authors:** Sergey V. Panin, Alexey A. Bogdanov, Alexander V. Eremin, Dmitry G. Buslovich, Vladislav O. Alexenko

**Affiliations:** 1Laboratory of Mechanics of Polymer Composite Materials, Institute of Strength Physics and Materials Science of Siberian Branch of Russian Academy of Sciences, 634055 Tomsk, Russia; ispmsbogdanov@gmail.com (A.A.B.); ave@ispms.ru (A.V.E.); buslovich@ispms.ru (D.G.B.); vl.aleksenko@mail.ru (V.O.A.); 2Department of Materials Science, Engineering School of Advanced Manufacturing Technologies, National Research Tomsk Polytechnic University, 634050 Tomsk, Russia

**Keywords:** polyimide, milled carbon fibers (MCF), polytetrafluoroethylene (PTFE), polymer composite, fatigue analysis, mechanical testing, hysteresis loop

## Abstract

The fatigue properties of neat polyimide and the “polyimide + 10 wt.% milled carbon fibers + 10 wt.% polytetrafluoroethylene” composite were investigated under various cyclic loading conditions. In contrast to most of the reported studies, constructing of hysteresis loops was performed through the strain assessment using the non-contact 2D Digital Image Correlation method. The accumulation of cyclic damage was analyzed by calculating parameters of mechanical hysteresis loops. They were: (i) the energy losses (hysteresis loop area), (ii) the dynamic modulus (proportional to the compliance/stiffness of the material) and (iii) the damping capacity (calculated through the dissipated and total mechanical energies). On average, the reduction in energy losses reached 10–18% at the onset of fracture, whereas the modulus variation did not exceed 2.5% of the nominal value. The energy losses decreased from 20 down to 18 J/m^3^ (10%) for the composite, whereas they reduced from 30 down to 25 J/m^3^ (17%) for neat PI in the low-cycle fatigue mode. For high-cycle fatigue, energy losses decreased from 10 to 9 J/m^3^ (10%) and from 17 to 14 J/m^3^ (18%) for neat PI and composite, respectively. For this reason, the changes of the energy losses due to hysteresis are of prospects for the characterization of both neat PI and the reinforced PI-based composites.

## 1. Introduction

Polyimides (PI) are high performance polymers (HPP), which possess enhanced thermal-oxidation stability, elastic modulus, radiation and chemical resistance, as well as maintain high mechanical properties over a wide temperature range [1,2,3]. These polymers are used for manufacturing components of highly loaded friction units [4,5]. However, high coefficient of friction and low wear resistance are characteristic features of the PI.

One way to resolve this problem is loading PI with solid lubricants that reduce its coefficient of friction [6,7,8,9] (for example, polytetrafluoroethylene (PTFE) [10]). However, this is accompanied by a decrease in the strength properties of the composites in most cases [11,12,13,14], which can be compensated by addition of reinforcing fibers [15]. A key aspect in the design of reinforced polymer composites is ensuring adhesion between the polymer matrix and fillers [16]. This challenge could be overcome by treating fibers with a coupling agent [17]. In our previous study [18], the mechanical and tribological properties of antifriction PI-based composites, filled with PTFE and milled carbon fibers (MCF), were studied. It was shown that wear rate of the “PI + PTFE + MCF” composite decreased by ∼310 times for the metal-polymer tribological contact and ∼285 times for the ceramic-polymer one (in comparison with that for neat PI).

Operation of polymer composites as a part of friction unit implies both contact pressure and wear, as well as various types of periodic loads. This imposes increased requirements on their mechanical properties. However, a few sets of data have been published so far on the effect of structure of antifriction HPP-based particulate composites (in particular, the PI-based ones) on their fatigue and static strength properties.

Analysis of the published data suggests that aspects of fatigue resistance were investigated primarily for neat polymers or composites loaded with continuous reinforcing fibers (fabric). Shrestha et al. [19,20] discussed the fatigue behavior, relaxation effects, the influence of mean strains per cycle, as well as the fracture patterns under cyclic loads for neat thermoplastic polymers. Studies of composites loaded with fibers were also discussed in the relevant literature. The effect of the volume fraction of short carbon fibers on mechanical properties, damage mechanisms, and fatigue life of PI-based composites were presented by Garcia et al. in [21]. It was shown that reinforcement with oriented short fibers can ensure mechanical and fatigue properties comparative with those for composites loaded with long [22] and continuous ones [23].

Both neat PI and short-fiber-reinforced polymer systems exhibit different fracture mechanisms under cyclic loading. Plastic strains prevail at high level of applied stresses thus leading to low durability. In doing so, the external load favors plastic flow development in the polymer matrix. Its localization gives rise to failure. At the lower level of applied stresses, the number of cycles upon failure increases due to different fracture mechanism. Structural defects are the sites of crack initiation. The latter develop under the applied loads until one of them reaches a critical size and fracture occurs [24].

Area of hysteresis loops [19], as well as dynamic modulus [25], are employed in the literature for accurate analysis of deformation processes that develop in structural materials under cyclic loading. Baxter [26] conducted a comprehensive study on the application of the load-stroke hysteresis technique. In doing so, a well-defined technique was used for the assessment of hysteresis loop parameters in non-destructive manner. However, moduli measurements were not performed, and hysteresis loop parameters were measured using strain gauging. Ruggles-Wrenn et al. [27] noted that variation of both the area, and the shape of the hysteresis loops indicate on the damage accumulation development.

Nevertheless, changes in the areas of hysteresis loops were not used to estimate and predict the residual life under fatigue tests, since the areas were considered to be a function of the cyclic load frequency [28]. McKeen [29] reported that the hysteresis loop shape and area stabilized upon 10% of the total durability and remained constant after. In addition, the loss of viscoelastic energy due to the hysteresis included energy dissipation for heat production and structural changes under cyclic loading [30].

Most models describing changes in hysteresis behavior under cyclic loading were based on phenomenological approaches and were developed for PI-based laminate composites. Damage mechanisms were not considered in these methods, although relevant damage indicators such as fatigue stiffness, fatigue strength, residual strength, etc., were used to assess their levels. The damage rate was a function of many factors, including applied cyclic loads, number of fatigue cycles, frequency, and environmental conditions, etc. [31]. Benaarbia et al. [32] showed that mechanical and thermodynamic cyclic stability were not observed in the fatigue tests. This was related to nonuniform strain distributions over bulk samples, resulting in uneven energy dissipation (hysteresis).

Decreasing of the hysteresis loop area under fatigue was revealed in PI-based composites tested in tension mode (R = 0.1) [33]. On the other hand, loops with increasing area were evident for these composites tested in alternative mode (R = −1). In doing so, hysteresis loop area is an important parameter that enables us to study both linear and nonlinear behaviors of the particulate composites, as well as to assess their damage degree under cyclic loading [34].

The modulus increased at the initial stage of the fatigue tests [35]. However, it remained constant for most of the operation life with a certain drop upon fracture. In contrast, energy dissipation was reduced at the beginning of the tests. Then, its reduction rate slowed down. Finally, the energy dissipation tended to enlarge, being concluded by the sample fracture.

Sutton et al. [36] investigated the fatigue behavior of PI-based composites using the concept of damage formation. This theory considers a fatigue crack and the surrounding processing zone at its tip as a thermodynamic object, where the material damage degree is associated with the system entropy and the energy dissipation rate for the initiation of a new surface. The rate of energy dissipation is defined as the difference in hysteresis energy of a specimen with and without cracks. The modified crack layer (MCL) theory was used to estimate the specific damage energy. The latter was indicative of the fatigue strength of the material. It was found that MCL analysis, based on increasing energy during damage formation, could reveal changes in fracture strength due to loading with various fillers. It was shown that polymer loaded with PTFE particles had a higher fracture toughness than fiber-reinforced composites. In this case, hysteresis energy was associated with crack growth, whereas the dissipation energy increased with fatigue crack growth. Loading polymer with three or more fillers exerted an ambiguous effect on the structure (including its heterogeneity) and strength properties. In addition, the aspects of fatigue resistance of high-performance polymer particulate composites were studied only fragmentarily so far.

The aim of the study was investigation of the fatigue behavior of three-component antifriction PI-based composite loaded with MCF and solid lubricant PTFE particles [18]. Unlike most reported results, stress assessment for constructing hysteresis loops was conducted using the non-contact 2D DIC method. The accumulation of cyclic damage was analyzed by calculating parameters of mechanical hysteresis loops. They were the hysteresis losses, the dynamic modulus (proportional to the compliance/stiffness of the material) and the damping capacity (calculated through the dissipated and total mechanical energies). The novelty of the study is related to establishing the correlation between structure and fatigue behavior of PI and an advanced three-component PI composite in terms of mechanical hysteresis loop parameters.

## 2. Materials and Methods

Four types of samples were subjected to static tensile tests. The first one was neat PI. The “Solver PI-Powder 1600” powder (Solver Polyimide Co., Jiande, China) with an average particle size of 16 μm was employed. The second one was a PI-based composite reinforced with milted carbon fibers (MCF) The “Tenax” MCF (Teijin Carbon Europe GmbH, Wuppertal, Germany) had a length of 200 μm and an aspect ratio of 30. The third one was the PI-based composite loaded with PTFE particles. The “Fluralit” powder of PTFE (Fluralit synthesis, Moscow, Russia) with an average particle size of 3 μm was used as a solid-lubricant filler. Both PTFE and MCF contents were 10 wt.% (for each component). The last one was three-component PI-based composite with 10 wt.% PTFE and 10 wt.% MCF; the material is referred as the “PI/PTFE/MCF” composite hereinafter. Fatigue properties were investigated for neat PI and “PI/PTFE/MCF”.

The polymer powders and fillers were mixed by dispersing the suspension components in the alcohol using an ultrasonic cleaner “PSB Gals 1335 05” (“PSB Gals” Ultrasonic Equipment Centre, Moscow, Russia). Processing time was 3 min; the generator frequency was 22 kHz. After mixing, a suspension of the components was dried in a forced-ventilation oven at 120 °C for three hours. The PI-based composite was produced by hot pressing at a pressure of 15 MPa and a temperature of 370 °C with a subsequent cooling rate of 2 °C/min. The basic plates were ‘70 × 60 × 10’ mm^3^ in size (Figure 1) and the samples were cut using a milling machine.

Structural studies were conducted using mechanically fractured notched samples after exposure to liquid nitrogen (197 °C) for one hour. The fractured surfaces were used to observe the distribution of fillers and the overall composite structure. The size of the samples was ‘55 × 10 × 10’ mm^3^. The notches with a depth of 2 mm and a curvature radius of 0.25 ± 0.01 mm were formed using a double-tooth disk cutter with diamond cutting edges and a ‘GT-7016-A3’ V-notch sampling machine (Gotech Testing Machines, Taichung City, Taiwan). Copper films about 10 nm thick were deposited on the fracture surfaces using a “JEOL JEE-420” vacuum evaporator (JEOL USA, Inc., Peabody, MA, USA). A “LEO EVO 50” scanning electron microscope (Carl Zeiss, Oberkochen, Germany) was employed at an accelerating voltage of 20 kV.

The dog-shaped specimens for the static and cyclic tests were 64 mm in length. Their gauge length was 10 mm long with a cross-section of ‘3.2 × 3.2’ mm^3^. The required surface quality was ensured by further polishing with sandpapers of various grit sizes up to 1000 grit. The tests were carried out using a “Biss Nano 15 kN” servo-hydraulic testing machine (Bangalore Integrated System Solutions Pvt Ltd., Bengaluru, India), (Figure 2).

Non-contact strains measurement during the static tests was carried out by the Digital Image Correlation (DIC) method [37] using a “VIC 2D” facility (Correlated Solutions Inc., Columbia, SC, USA).

The specimen temperature was assessed on their surfaces with a ‘Melexis MLX90614’ digital non-contact infrared thermometer (Melexis, Belgium). To eliminate its rising and affecting the fatigue properties, a cycle frequency of 1 Hz was preset, at which the deviation of the specimen temperature from the room level did not exceed 2 °C during the tests.

The static tensile tests were performed according to ASTM D 638 standard in order to determine the key mechanical properties: (i) tensile strength, (ii) elongation at break, (iii) elastic modulus, and (iv) yield strength (0.1%) at room temperature. The crosshead movement speed was 0.24 mm/min.

The cyclic tensile tests were carried out according to ASTM E606 standard in the displacement control mode (strain ratio R = 0). The cycle shape was triangular; the frequency was 1 Hz. To assess the deformation behavior and damage accumulation of the polymers, several parameters based on the analysis of mechanical hysteresis loops were calculated.


1.The dynamic modulus ($E_{dyn}$) was determined as a fitting line slope between two points of a hysteresis loop. The dynamic modulus was calculated by the ratio of the stress range over the strain one (Equation (1) and illustrated in Figure 1):(1)$$E_{dyn}=\left|E^{*}\right|=\frac{\Delta \sigma }{\Delta \varepsilon }=\frac{\sigma_{max}-\sigma_{min}}{\varepsilon_{max}-\varepsilon_{min}}$$
where *E_dyn_* is the dynamic modulus which is equal to the absolute value of the complex modulus (|*E*^∗^|); the latter characterizes the viscoelastic behavior of polymer materials. Due to viscous nature, a strain rate of viscoelastic materials depends on the time and exhibits hysteresis variation pattern.2.The second parameter was a hysteresis loop area (*Energy loss*), which corresponds to energy loss at each cycle (Figure 1). The energy dissipation is associated with heating as well as structure rearrangements [38]. The hysteresis loop area is a measure of energy losses under cyclic loading. In this paper, the *E_dyn_* and energy loss were employed to characterize the structural state of the cyclically loaded neat PI and the PI-based composite.3.The strain energy was calculated as the area under the loading curve from its beginning to the point of reaching the maximum value (Figure 3).4.To assess the damping capacity of the studied materials, the ratio of energy loss over the total strain energy in a cycle was determined. The relative damping value was estimated as:
(2)$$\psi =100\;\frac{Energy\;loss}{Strain\;Energy}\%,$$
where *Energy loss* is the work of internal forces upon a full cycle of variable loads, that was defined as the hysteresis loop area; *Strain energy* is the total energy in a cycle.


## 3. Results

### 3.1. Structural Study

Figure 2 shows SEM micrographs of neat PI and the “PI/PTFE/MCF” composite. A homogeneous structure was evident in neat PI (Figure 4).

It consisted of structural elements in the form of rounded “cells” with a characteristic dimension of 15 µm, which were comparable to the particle diameters of the original powder. Loading PI with PTFE and MCF prevented the molten polymer from spreading uniformly during compression molding. This resulted in porosity and heterogeneity (Figure 4b). In general, the molten polymer flowed satisfactorily around the fibers, hence an increase in strength properties could be expected. However, numerous finely dispersed small PTFE inclusions, which did not adhere to the PI matrix, resulted in the “loose” composite structure that facilitated reducing strain properties (will be described in the next section). The residual porosity of the PI-based composite obtained from hot pressing of the powder mix did not exceed 1.0%. Thus, a large number of discontinuities were formed (Figure 4b), which were stress raisers and potential sites for damage initiation (microcracks) in the PI-based composite.

### 3.2. Static Loading

Figure 5 presents stress–strain diagrams for neat PI and the “PI/PTFE/MCF” composite. In addition, the graphs of the two-component “PI/CF” and “PI/PTFE” composites are presented for comparison. At least five specimens of each type of material were tested. The most important mechanical characteristics are summarized in Table 1. Loading PI with MCF significantly enhanced the elastic modulus. In turn, the PTFE addition into PI gave rise to a decrease in both strength and the elastic modulus. The yield point of the composite “PI/PTFE/MCF” increased by 1.4 times in comparison with neat PI; its elastic modulus enhanced by 2.2 times but elongation at break decreased by 3.3 times.

The inelastic strain *ε_f_* was predictably lower for the “PI/PTFE/MCF” composite than that for neat PI (both in absolute and relative terms). This was directly related to the ability to re-arrange the inner structure of the polymer, which was significantly suppressed by loading with MCF and partially by PTFE particles.

As it was shown in the previous paper [18], the application of the two-component “PI/MCF” composite filled with carbon fibers in the steel-polymer tribological contact is limited by the accelerated counterpart wear. The “PI/PTFE” sample possessed the advanced anti-friction characteristics, but it has low mechanical properties. The range of industrial applications of polymer composites depends, inter alia, on their yield strength and rigidity. In this regard, the triple “PI/PTFE/MCF” composite is characterized by the greatest elastic modulus and yield strength among the studied ones. For this reason, the fatigue properties of the ‘PI/PTFE/MCF’ composite and neat PI were compared below.

### 3.3. Fatigue Tests

The results of the fatigue tests are shown in Table 2. The latter summarizes the parameters for the first loading cycle (the maximum strain *ε*_max_ and stress ***σ***_max_ levels), as well as the number of cycles prior to failure. The tests were carried out in the displacement (strain) control mode. The number of specimens tested under three different strain levels for each type of material was equal to seven.

Figure 6 shows fatigue curves for neat PI and the “PI/PTFE/MCF” composite in two types of coordinates: (a) “strain amplitude vs. cycles to failure” and (b) “stress amplitude of the first cycle vs. cycles to failure”. Thin horizontal lines show the yield strength levels for each of the materials, which are essentially the border between high-cycle fatigue (HCF) and low-cycle fatigue (LCF). “PI/PTFE/MCF” composite curve was located lower in the “strain amplitude vs. cycles to failure” coordinates (Figure 4a), whereas it was higher in the second case (Figure 4b) due to the higher yield point and elastic modulus.

The decrease in durability at the (Δ*ε*/2) equal strains of the PI/PTFE/MCF’ composite was explained by its lower *ε* elongation at break. In turn, the higher both elastic modulus and yield strength resulted in the greater durability at the same (Δ***σ***/2) stress level in a cycle. In the LCF region (10^3^–10^4^ cycles), the difference in the amplitude (Δ***σ***/2) of the withstand stresses was about 10 MPa that corresponded to the variations of the yield strength of the materials. Thus, the increase in fatigue life for the “PI/PTFE/MCF” composite was more pronounced in the LCF range due to a smaller proportion of plastic strains. The stress fatigue limit was higher for the “PI/PTFE/MCF” composite, which was predictable and related to its higher yield/strength level.

Figure 7a presents the fatigue curves obtained by recalculating the applied strain relative to the strain at the yield point. The curves are located close to each other. For each one, three characteristic sections could be distinguished. In the first section, a significant increase in durability was accompanied by a slight decrease in the cyclic load amplitude, which corresponded to the LCF mode (up to 10^4^ cycles). The second was a transitional one between the LCF and HCF modes; its middle part corresponded to the yield point. This section was characterized by a slight increase in durability with a decrease in the cyclic load amplitude. The third section began at ≈10^5^ cycles and had a gentle slope of the curves that corresponded to the HCF mode. In this section, even small changes in the amplitude levels caused very significant variations in durability. The fatigue curves (Figure 7b), drawn after recalculating the applied stresses relative to the yield point, were also similar but located farther apart.

Note that neat PI was capable to relax stresses due to its higher plasticity and the ability of flexible conformation of the polymer molecular structure. On the other hand, the “PI/PTFE/MCF” composite possessed higher rigidity and had more barriers for the propagation of microcracks, primarily at the “matrix–fiber” boundaries. However, damages occurred more easily at such boundaries and resulted in crack initiation.

### 3.4. Fractographic Studies

Figure 8 shows SEM micrographs of the fracture surfaces of the neat PI after static and cyclic tests. The neat PI fracture surfaces after the static tensile tests are shown in Figure 8 (left column). A main crack nucleated on a defect near the sample surface (Figure 8a). It is as well seen that the fracture surface consisted of two main zones. A “cellular” relief pattern, shown in Figure 6b, corresponded to the first stage of its propagation. This was followed by the final rupture zone, which possessed traces of normal peel accompanied by local stretching of the polymer (Figure 8c). In this regard, a heterogeneous ridge relief with large structural elements was observed on the fracture surfaces.

For LCF at *ε*/*ε*_0.1_ = 1.3, the fracture surface exhibited three characteristic zones (Figure 8, middle column): ones of stable and accelerated crack propagation (Figure 8d,e), as well as a final rupture zone (Figure 8f). The stable crack propagation zone was characterized by typical round shaped structural elements, as well as fatigue striations which had been formed when crack transited to accelerated propagation (Figure 8e). The relief formed in the final rupture zone was associated with the development of plastic strains and was accompanied by local stretching of the polymer upon the crack propagation by the normal opening mechanism. The final rupture zone had a similar appearance with that after the static test, but possessed a less-developed fracture surface relief.

For HCF at *ε*/*ε*_0.1_ = 0.7, the specimens withstood more than 10^5^ cycles at the load amplitude below the yield point. For this reason, the fracture surface appeared quite smooth and uniform. As in the LCF case, three fracture zones were distinguished: the stable and accelerated crack propagation ones (Figure 8g,h), and the final rupture zone (Figure 8h,i). In the stable crack propagation zone, the relief was similar to that for the LCF mode. In the unstable crack propagation zone, which was characterized by the rather rapid process development, the fractographic relief as a whole followed the pattern of neat PI supermolecular structure (both in appearance and in the size of structural elements).

SEM micrographs of the “PI/PTFE/MCF” composite fracture surfaces are presented in Figure 9. For the ones after the static tests (Figure 9, left column), the relief was heterogeneous in height. On the other hand, the MCF distribution was uniform, whereas PTFE particles were not agglomerated. The high degree of relief heterogeneity does not allow to find the zone of crack initiation and propagation in the composite.

In the LCF mode (Figure 9, middle column), the fracture surfaces were also homogeneous: regions with a smoother topography were adjacent to rougher ones, which showed signs of fracture by the normal opening mechanisms with a short duration of this process.

For the HCF mode, the macroscopic fracture (Figure 9, right column) appeared to be the smoothest due to repeated cyclic loads and the gradual accumulation of damages. An important revealed feature was the presence of regions “depleted” in MCF on the fracture surface (Figure 9g–i). In addition, traces of pulling out fibers were observed, which were oriented parallel to the fracture surface. In such areas, longitudinal “stripes” were presented. Some of them were due to the separation of fibers from the matrix, but others had the fatigue nature of origin (Figure 9i). Although the preliminary air annealing of carbon fibers promoted a minor increase in interfacial adhesion (which leads to higher mechanical properties), SEM micrographs showed that a strong interfacial bonding did not occur.

In addition, despite these variations, the general fracture surface patterns of the “PI/PTFE/MCF” composite are to be considered similar for all three cases. The reason for this fact was its reinforcement with MCF, which restrained pronounced plastic strains and the development of structural rearrangements in the polymer matrix.

Thereby, the conventional fractography analysis did not provide enough information to interpret the differences in the strain behavior and fatigue life of the studied materials due to the small contribution of the crack propagation stage to fatigue life. In this regard, further analysis of the parameters of mechanical hysteresis loops was carried out in order to identify their relationship with the fatigue fracture resistance. In particular, two parameters were used: the dynamic modulus and the hysteresis loop area.

## 4. Analysis of the Mechanical Hysteresis Loops

The kinetics of energy losses upon the fatigue tests in both LCF and HCF modes are presented in Figure 8. These graphs showed that LCF and HCF modes differed by a factor of two in hysteresis energy losses. Losses per cycle were ~20 J/m^3^ for neat PI in the LCF mode and ~10 J/m^3^ in HCF, whereas these levels were ~26 and ~13 J/m^3^ for the composite (Figure 10a; curves 1, 3 and Figure 10b; curves 1, 3). The composite possessed a 30% higher energy loss per cycle compared to neat PI. In the LCF mode, similar trends towards energy loss were evident for both materials with enlarging the number of cycles (by ~3 J/m^3^ for neat PI and by ~5 J/m^3^ for the composite). Energy losses almost did not change in the HCF mode, thus the fitting straight lines on the graphs, have nearly zero inclination angles to the horizontal axis.

The *E_dyn_* modulus was generally lower at the LCF mode (*ε*/*ε*_0.1_ = 1.3), which was associated with the possibility of inelastic strain development. In this case, the *E_dyn_* modulus variation was three times greater than that in the HCF mode. For the composite, the *E_dyn_* modulus decreased by ~60 MPa in the LCF mode, whereas by ~20 MPa in the HCF (Figure 10a; curve 2 and Figure 10b; curve 2). In turn, neat PI showed lower *E_dyn_* rising: by ~30 MPa in the LCF mode and by ~10 MPa in the HCF (Figure 10a; curve 4 and Figure 10b; curve 4). Thereby, the ranges of both Energy loss and *E_dyn_* levels were significantly reduced when transiting from the LCF to the HCF mode due to the lowering intensity of ongoing structural changes. This fact could be interpreted as a decrease in the sensitivity of these parameters to the development of strain processes, including the accumulation of scattered damages. The scatter of both Energy loss and *E_dyn_* values reflected the instability of the damage development process in the bulk material.

The decrease in the Energy loss parameter on the graphs could be due to the action of two mechanisms: the reorientation of macromolecules in the polymer matrix and the development of defects at the interface between the composite’s components. However, these mechanisms affected the dynamic modulus in different ways. In particular, the reorientation of macromolecules increased the stiffness, i.e., dynamic modulus, whereas the accumulation of defects reduced it. The obtained results did not enable to unambiguously interpret the reason for the decrease in the Energy loss parameter as the defects developed. We suggest that finely dispersed PTFE inclusions could play an important role here, which contributed to the effective stress relaxation in the composite without disturbing its continuity. However, the slope of the curves on the Energy loss and *E_dyn_* plots could be an informative parameter for characterizing the fatigue properties and kinetics of both neat PI and its triple composite.

To assess the damping capacity (ψ) of the investigated materials, the ratio of the energy (hysteresis) losses over the total strain energy in a cycle was determined (Figure 11). Damping capacity was found to be 3 and 5% of the total strain energy for neat PI in both LCF and HCF modes (Figure 11a,b). For the composite, this parameter was two times higher (8 and 11%, correspondently). In fact, damping capacity was constant throughout the cyclic tests for each of the studied materials. The quantitative results of the energy analysis are presented in Table 3.

The increase in the damping capacity from ~3% for neat PI up to ~8% for the composite caused doubling of its fatigue durability in the LCF mode. In HCF mode, the damping capacity was changed from 5% to 11% for neat PI and the composite, respectively; in doing so, the lifetime of the composite was enhanced by 50%. In this way, a correlation was observed between the energy losses and the cyclic durability. The obtained results showed that the simultaneous reduction in energy losses and the enlargement of the *E_dyn_* modulus under cyclic loads was characteristic of neat PI.

## 5. Discussion

The authors suggest the following interpretation of the obtained results. Cyclic fracture toughness is related to the damping capacity of the materials, since it is proportional to strain energy dissipation by converting into heat (instead of developing of damages). Upon mechanical loading, neat PI (with the linear macromolecular structure) is characterized by a predominant reorientation of macrochains along the load direction. This stimulates the increase in the elastic properties (manifested in the *E_dyn_* growth), but reduces the cyclic fracture toughness (which was reflected through the reduction in the Energy loss). The critical state (typically interpreted as the exhaustion of the relaxation capacity) could be considered the moment, when the crucial fracture energy is below the strain energy. This corresponds to the initiation and subsequent propagation of a crack.

The above reasoning enables to draw the following conclusion. With an equal level of applied cyclic loads, the higher damping capacity provides the greater durability. It follows that the higher energy losses for the composite reflect its greater damping properties, which result in the increase in its fatigue life compared to neat PI.

When discussing the results of the cyclic tests, the reinforcing role of the MCF was mainly mentioned. However, the molecular structure of the matrix was also modified when loading with 10 wt.% PTFE. It is obvious that both the decrease in the dynamic modulus for the composite and its enhancement for the neat PI with the increase in the number of cycles were due to the variations of their structures, which caused the different deformation mechanisms. The reinforced composite possessed the heterogeneous structure in which the presence of fine and soft PTFE particles could lead to an increase in its damping capacity, thereby improving the durability. This was due to the fact that a greater amount of deformation energy was dissipated in the form of heat, but not for the development of damages. However, some strength properties were deteriorated in this case. On the other hand, loading with MCF restrained the deformation development, which enhanced the mechanical properties.

Thus, simultaneous filling with MCF and PTFE resulted in the fact that the composite, having the same static strength, withstood more cycles than neat PI at the same loading level.

For the composite, the decrease in the Energy Loss level under cyclic loads was caused by its filling with MCF. The rearrangement of the polymer structure with the predominant reorientation of its elements along the load direction occurred due to the conformation of macromolecules. These processes developed several times less intensively in the HCF mode. In the LCF case, the molecular structure of the composite changed to a lesser extent compared to neat PI. Upon cyclic loading, the deformation processes initiated and developed more actively at the composite’s internal interfaces. Since the deformation of high-strength MCFs was elastic only, the development of the hysteresis effects, which manifested themselves in the form of the corresponding loops, was associated with the strain processes in the composite polymer matrix alone.

In conclusion, the authors would like to draw attention to the role of adhesion in the deformation behavior of the composite. SEM micrographs of the three-component composite have shown that the level of adhesion between the matrix and both fillers was rather low. This imposed a limitation on the contribution of the reinforcing (short fibers) and solid lubricant PTFE additives to the mechanical (strength) properties. Static tensile test results, previously published by the authors [17], have shown that PI possessed a tensile strength of 110.7 MPa. The subsequent loading with the PTFE particles significantly reduced this value down to 77.7 MPa, whereas the addition of annealed carbon fibers (as the third component) gave rise to the increase in the ultimate strength up to 96.4 MPa. Thus, despite the low adhesion between the fibers and the PI matrix, the presence of reinforcing inclusions played a significant role due to the formation of extra interfaces. In addition, the interfacial adhesion should increase the yield and ultimate strength. In doing so, this should prevent the initiation and development of damages, thereby increasing the fatigue life.

## 6. Conclusions

The fatigue properties of neat PI and the “PI/PTFE/MCF” composite were investigated under various cyclic loads. The analysis of the fracture mechanisms was carried out considering the initial effect of the materials’ structure. Based on the obtained results, the following conclusions can be drawn.

Comparison of neat PI and the “PI/PTFE/MCF” composite showed that loading with 10 wt.% PTFE and 10 wt.% MCF resulted in the 2.2-fold increase in modulus, the 3.3-fold decrease in elongation at break, and the increase in the yield strength by 1.4 times for the “PI/PTFE/MCF” composite. The tensile strength did not change, hence the hardening MCF contribution was “compensated” by the softening effect caused by loading with PTFE.

The results of the fatigue tests showed that the “PI + PTFE + MCF” composite could withstand stresses higher than that for neat PI for the same number of cycles to failure. Under the conditions of equal stresses, the composite could withstand an order of magnitude more cycles to failure than neat PI. Thus, the fatigue characteristics of the materials could be very different for the same static strength. In contrast with the neat polymer, the lifetime of the composite was enhanced by 100% in the LCF mode, whereas it has been just 50% larger in the HCF one.

Under cyclic loads, the polymer was characterized by the increase in the modulus value. It was associated with the development of plastic strains in neat PI or in the PI matrix by the strain-induced structure rearrangement. During the cyclic tests, the *E_dyn_* modulus decreased by 60 MPa (2.5% of the initial value) in the LCF mode for the composite, whereas by 20 MPa (0.8%) in the HCF one. In turn, neat PI showed lower *E_dyn_* rising: by 30 MPa (2.4%) in the LCF mode and by ~10 MPa (0.8%) in the HCF. The variations of the dynamic modulus did not enable to clearly distinguish the damage accumulation mechanisms because of the negligible decrease in its level.

It was shown that the value of energy losses due to mechanical hysteresis was an informative measure of the mechanical state of cyclically deformed polymers and the reinforced PI-based composites. On average, the decrease in energy losses reached 10–18% by the onset of fracture, whereas the change in the modulus did not exceed 2.5% of the nominal value. The energy losses decreased from 20 J/m^3^ to 18 J/m^3^ (10%) for the composite and from 30 to 25 J/m^3^ (17%) for neat PI in the LCF mode. In the HCF case, energy losses decreased from 10 to 9 (10%) and from 17 to 14 J/m^3^ (18%) for neat PI and the composite, respectively. For this reason, the changes in the energy losses due to hysteresis are promising for the characterization of both neat PI and the reinforced PI-based composites.

The damping capacity was suggested as an energy-analysis-based fatigue property. The increase in the damping capacity from ~3% for neat PI up to ~8% for the composite gave rise to the 100% increase in its fatigue durability in the LCF mode. In the HCF mode, damping capacity was varied from 5% to 11% for neat PI and the composite, respectively, whereas the lifetime of the composite was enhanced by 50%. The observed effect was attributed to the amount of deformation energy dissipated in the form of heat but not for the development of damages.

## Figures and Tables

**Figure 1 materials-15-04656-f001:**
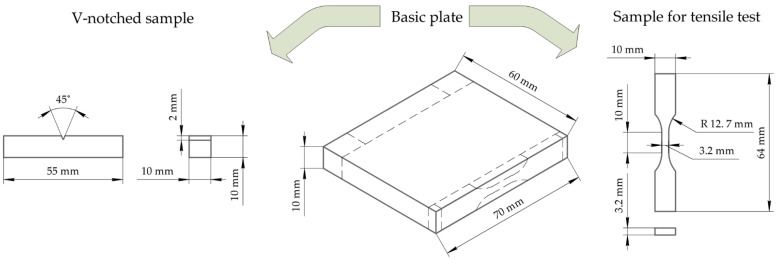
A schematic representation of the hot-pressed plate and cut test samples.

**Figure 2 materials-15-04656-f002:**
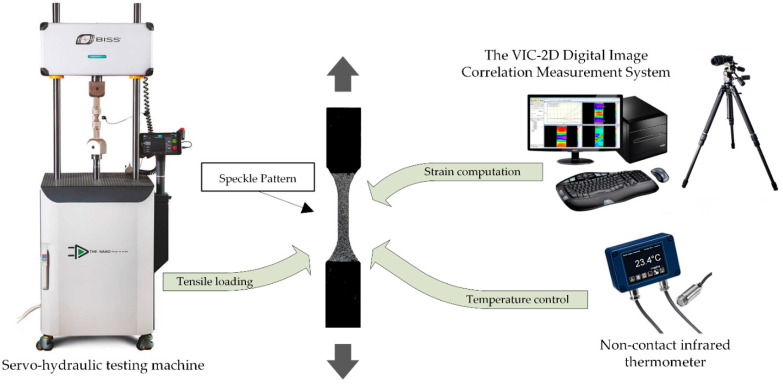
Experimental setup.

**Figure 3 materials-15-04656-f003:**
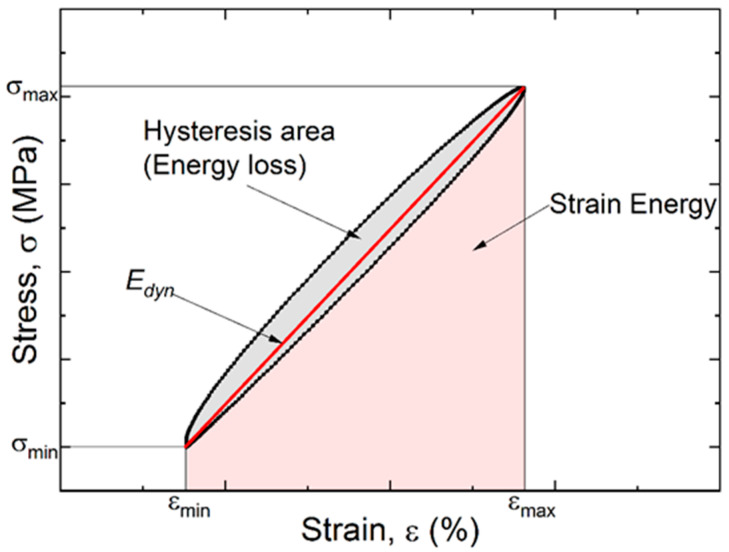
A schematic representation of the mechanical hysteresis loop and its parameters.

**Figure 4 materials-15-04656-f004:**
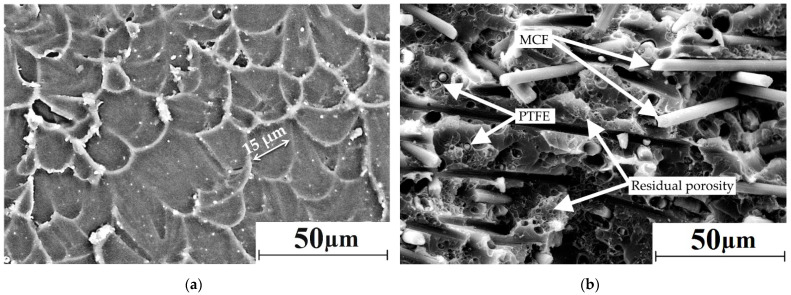
The SEM micrographs of the bulk structure: (**a**) neat PI; (**b**) the “PI/PTFE/MCF” composite.

**Figure 5 materials-15-04656-f005:**
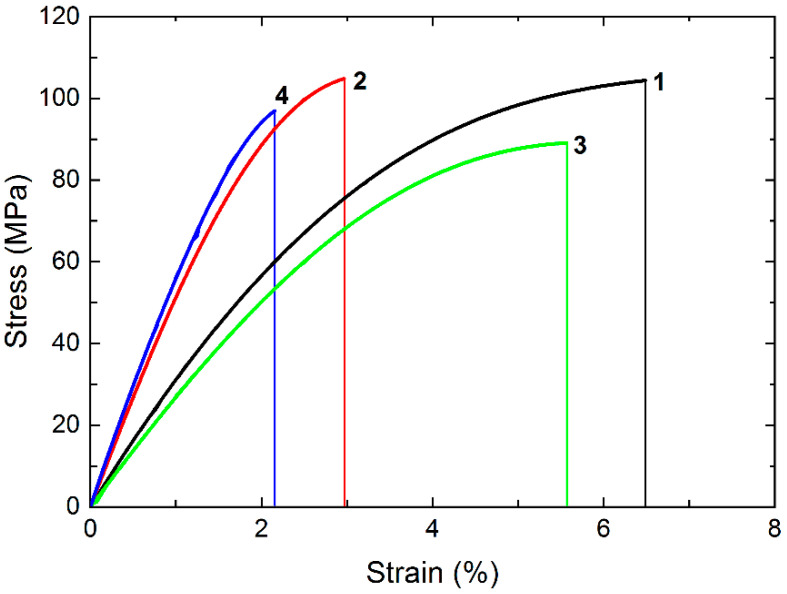
The stress–strain diagrams: 1 (black)—neat PI; 2 (red)—“PI/MCF” composite; 3 (green)—“PI/PTFE” composite; 4 (blue)—“PI/PTFE/MCF” composite.

**Figure 6 materials-15-04656-f006:**
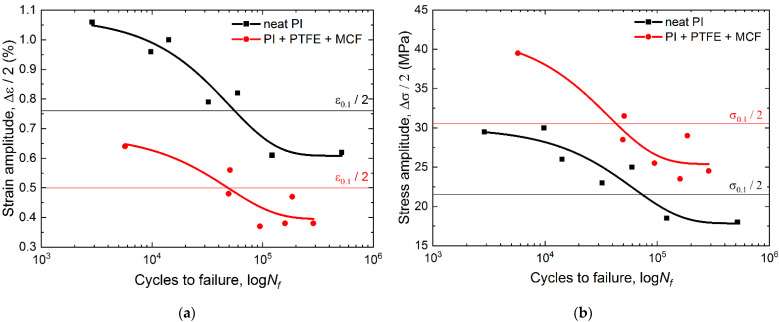
The fatigue curves for neat PI and the “PI/PTFE/MCF” composite according to the strain (**a**) and stress (**b**) amplitudes.

**Figure 7 materials-15-04656-f007:**
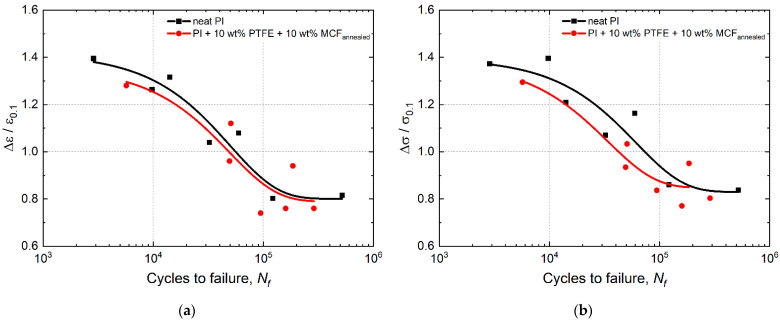
The strain (**a**) and stress (**b**) fatigue curves for neat PI and the “PI/PTFE/MCF” composite normalized over the yield point.

**Figure 8 materials-15-04656-f008:**
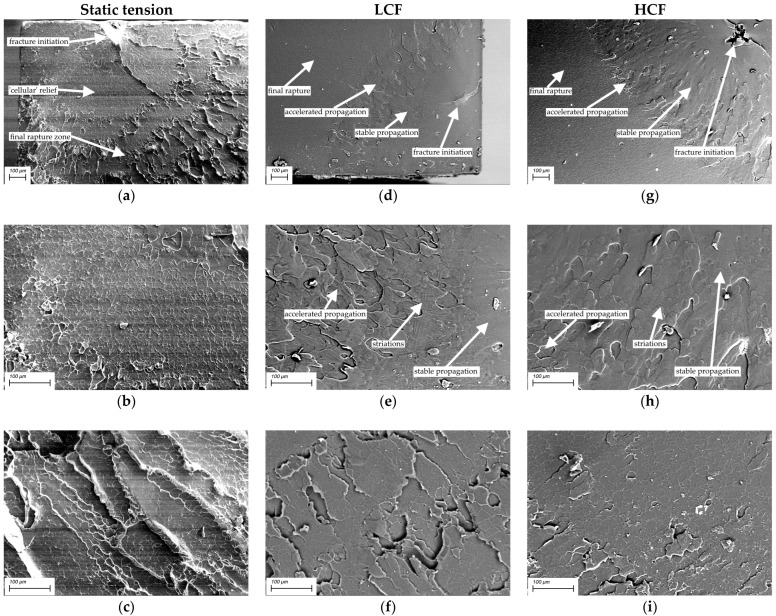
The SEM micrographs of the neat PI fracture surfaces: left column—under static tension, middle column—under LCF, right column—under HCF: (**a**) fracture initiation region, (**b**) “cellular” relief pattern, (**c**) final rupture zone, (**d**) fracture initiation region, (**e**) transition from stable crack propagation zone to its accelerated propagation one, (**f**) final rupture zone, (**g**) fracture initiation region, (**h**) transition from accelerated crack propagation zone to its final rupture one, (**i**) final rupture zone.

**Figure 9 materials-15-04656-f009:**
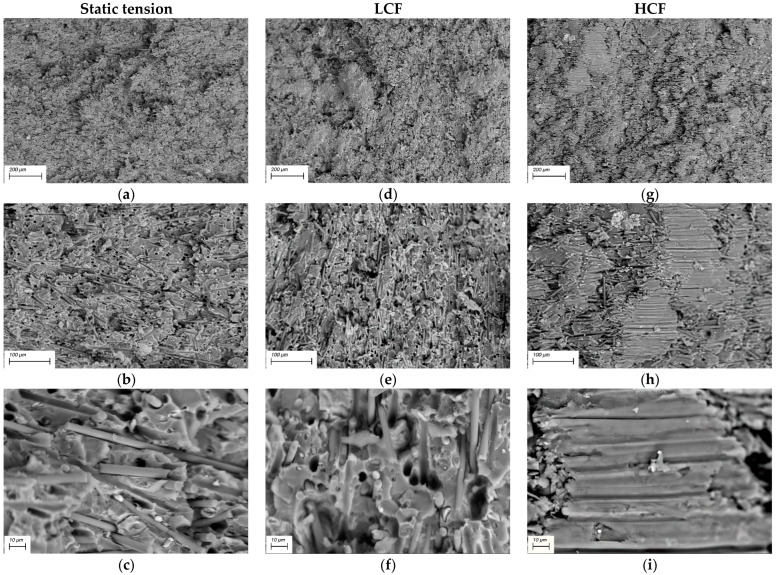
The SEM micrographs of the “PI/PTFE/MCF” composite fracture surfaces: left column—under static tension, middle column—under the LCF, right column—under the HCF: (**a**–**c**) high degree heterogeneity relief, (**d**–**f**) signs of fracture by the normal opening mechanisms, (**g**–**i**) MCF “depleted” regions in on the fracture surface.

**Figure 10 materials-15-04656-f010:**
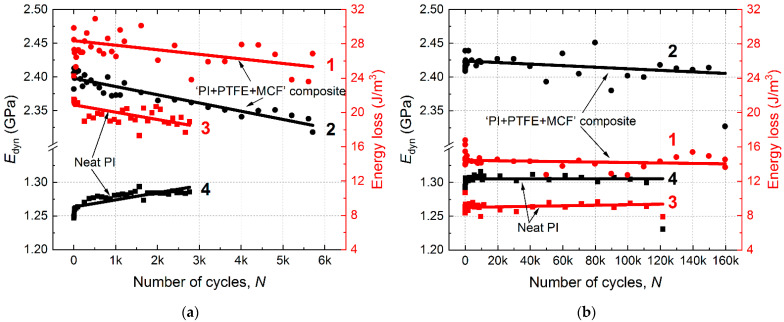
Changes in the dynamic modulus (black) and energy losses (red) vs. the number of cycles for neat PI and the “PI/PTFE/MCF” composite; LCF mode with ***σ*** = 1.3 of the yield point (**a**) and HCF mode with ***σ*** = 0.7 of the yield point (**b**). Curves designations are the following (1)—Energy loss for “PI/PTFE/MCF”; (2)—*E_dyn_* for “PI/PTFE/MCF”; (3)—Energy loss for neat PI; (4)—*E_dyn_* for neat PI.

**Figure 11 materials-15-04656-f011:**
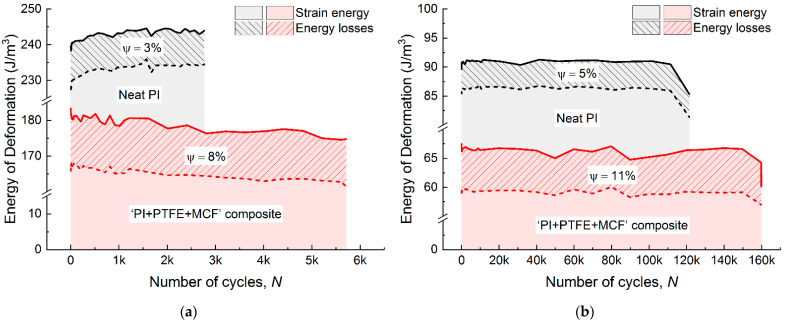
The total strain energy, energy losses and damping capacity for neat PI and the “PI/PTFE/MCF” composite; LCF mode with ***σ*** = 1.3 of the yield point (**a**) and HCF mode with ***σ*** = 0.7 of the yield point (**b**).

**Table 1 materials-15-04656-t001:** The results of the static tensile tests.

Material	*σ*_UTS_ (MPa)	*ε* (%)	*E* (GPa)	*σ*_0.1_ (MPa)	*ε*_0.1_ (%)	*ε_f_* (%) ^1^
Neat PI	104 ± 4	6.5 ± 0.4	3.08 ± 0.15	43 ± 3	1.52 ± 0.13	70
“PI/PTFE” composite	89 ± 5	5.6 ± 0.2	2.75 ± 0.18	42 ± 4	1.64 ± 0.09	71
“PI/MCF” composite	105 ± 4	3.0 ± 0.3	5.41 ± 0.21	63 ± 3	1.26 ± 0.11	58
“PI/PTFE/MCF” composite	97 ± 3	2.0 ± 0.2	6.9 ± 0.3	61 ± 4	1.00 ± 0.06	50

^1^*ε_f_* is the proportion (in percent) of inelastic (irreversible) strains in the total strains before fracture and found as *ε_f_* = (*ε* − *ε*_0.1_)/*ε* 100%.

**Table 2 materials-15-04656-t002:** The results of the fatigue tests for neat PI and the “PI/PTFE/MCF” composite.

*ε*/*ε*_0.1_	*ε*_max_ (%)	*σ*_max_ (MPa)	N*_f_*
Neat PI			
1.3	2.0	61 ± 6	9000 ± 4000
1.0	1.5	52 ± 3	50,000 ± 8000
0.7	1.1	41 ± 2	110,000 ± 10,000
“PI/PTFE/MCF” composite			
1.2	1.2	80 ± 6	6000 ± 3000
1.0	1.0	63 ± 4	50,000 ± 10,000
0.7	0.7	53 ± 3	130,000 ± 20,000

**Table 3 materials-15-04656-t003:** The energy parameters of fatigue tests for neat PI and the “PI/PTFE/MCF” composite.

		LCF	HCF
Strain Energy for the first cycle (Pa = J/m^3^)	Neat PI	228	85
	“PI + PTFE + MCF”	168	59
Energy loss for the first cycle (J/m^3^)	Neat PI	20	10
	“PI + PTFE + MCF”	30 (+50%)	17 (+70%)
Total Strain Energy (MJ/m^3^)	Neat PI	0.67	11.00
	“PI + PTFE + MCF”	1.02	10.57
Total Energy loss (MJ/m^3^)	Neat PI	0.02	0.53
	“PI + PTFE + MCF”	0.08	1.12
Damping capacity ψ (%)	Neat PI	3%	5%
	“PI + PTFE + MCF”	8%	11%
Fatigue life (Cycles ×10^3^)	Neat PI	3	120
	“PI + PTFE + MCF”	6 (+100%)	180 (+50%)

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
