# Peer review of "Estimating Low- and High-Cyclic Fatigue of Polyimide-CF-PTFE Composite through Variation of Mechanical Hysteresis Loops"

_materials, 2022, doi:10.3390/ma15134656_

Round 1

Reviewer 1 Report

1.  Pictures of your experimental samples will greatly enhance readers understanding of your work. (Lines 158, 163 and 170).

2.  Please include a picture of your experimental setup.  (Lines 173, 176, 178)

3.  For each of the quasi-static and fatigue tests, can you specify which material would be used for a specific purpose?

Author Response

Respectful reviewer,

many thanks for your hard job on careful reading and evaluation of the manuscript. All your comments are valuable and absolutely correct. We addressed them exactly when revises the manuscript. While making the corrections, the manuscript has undoubtedly been improved both in terms of clarity and discussion of the data presented.

Changes in text can be followed with the use of the “Track Changes” function.

  1.  Pictures of your experimental samples will greatly enhance readers understanding of your work. (Lines 158, 163 and 170).

- Thank you for this relevant comment. The geometry of the used samples and the scheme of their cutting have been added.

  1.  Please include a picture of your experimental setup.  (Lines 173, 176, 178)

- The photographs of the equipment used in the experiments have been added as well.

  1.  For each of the quasi-static and fatigue tests, can you specify which material would be used for a specific purpose?

- Neat polyimide is comparable to the PI/MUV/PTFE composite in terms of tensile strength; however, the elastic modulus and yield strength of the former are lower. Anyway, unfilled polyimide is of practical importance at operation conditions close to quasi-static loading ones. The PI/MUV/PTFE composite is recommended for the use under cyclic loading conditions due to its increased fatigue resistance.

Reviewer 2 Report

In this paper, the fatigue behavior of neat polyimide and polyimide composites has been systematically researched. However, this paper is not appropriate for publication now.

1. There are lots of language problems that need to be addressed. The whole manuscript should be carefully revised.

2. The abbreviations appearing for the first time should be written clearly in full.

3. The introduction part of this paper is not well organized. The aim is not clear, and the description of the existing research is very confusing.

4. Where does the crack initiation of the PI/PTFE/MCF composite occur? Surface or inside?

Author Response

Respectful reviewer,

many thanks for your hard job on careful reading and evaluation of the manuscript. All your comments are valuable and absolutely correct. We addressed them exactly when revised the manuscript. While making the corrections, the manuscript has undoubtedly been improved both in terms of clarity and discussion of the data presented.

Changes in text can be followed with the use of the “Track Changes” function.

  1. There are lots of language problems that need to be addressed. The whole manuscript should be carefully revised.

- We are sorry for making this impression. The manuscript has been duly edited and revised.

  1. The abbreviations appearing for the first time should be written clearly in full.

Abbreviations have been removed from the Abstract. Their descriptions have been added to the text.

  1. The introduction part of this paper is not well organized. The aim is not clear, and the description of the existing research is very confusing.

- Thank you for the relevant comment. We did our best to address this comment. The required changes have been made in the Introduction section.

  1. Where does the crack initiation of the PI/PTFE/MCF composite occur? Surface or inside?

Thank you for this relevant comment. The high degree of heterogeneity of fracture surface relief did not allow revealing the exact region of crack initiation in the composite. We suggest that crack initiated in the bulk composite due to large number of stress risers there.

Round 2

Reviewer 2 Report

The manuscript has been revised as required.